# Refining pairwise sequence alignments of membrane proteins by the incorporation of anchors

René Staritzbichler[1][☯]*, Edoardo Sarti[2,3][☯], Emily Yaklich[2][☯], Antoniya Aleksandrova[2], Marcus Stamm[4], Kamil Khafizov[5], Lucy R. Forrest[2]*

**1** ProteinFormatics Group, Institute of Biophysics and Medical Physics, University of Leipzig, Leipzig, Germany, **2** Computational Structural Biology Section, National Institutes of Neurological Disorders and Stroke, National Institutes of Health, Bethesda, MD, United States of America, **3** Laboratoire de Biologie Computationnelle et Quantitative, Institut de Biologie Paris Seine, Sorbonne Université, Paris, France, **4** Max Planck Institute of Biophysics, Frankfurt am Main, Germany, **5** Moscow Institute of Physics and Technology, National Research University, Moscow, Russia

☯ These authors contributed equally to this work.
* rene.staritzbichler@medizin.uni-leipzig.de (RS); lucy.forrest@nih.gov (LRF)

**Data Availability Statement:** The data associated with the manuscript is now available on zenodo: with the following DOI: 10.5281/zenodo.4016927.

## Abstract

The alignment of primary sequences is a fundamental step in the analysis of protein structure, function, and evolution, and in the generation of homology-based models. Integral membrane proteins pose a significant challenge for such sequence alignment approaches, because their evolutionary relationships can be very remote, and because a high content of hydrophobic amino acids reduces their complexity. Frequently, biochemical or biophysical data is available that informs the optimum alignment, for example, indicating specific positions that share common functional or structural roles. Currently, if those positions are not correctly matched by a standard pairwise sequence alignment procedure, the incorporation of such information into the alignment is typically addressed in an ad hoc manner, with manual adjustments. However, such modifications are problematic because they reduce the robustness and reproducibility of the aligned regions either side of the newly matched positions. Previous studies have introduced restraints as a means to impose the matching of positions during sequence alignments, originally in the context of genome assembly. Here we introduce position restraints, or "anchors" as a feature in our alignment tool AlignMe, providing an aid to pairwise global sequence alignment of alpha-helical membrane proteins. Applying this approach to realistic scenarios involving distantly-related and low complexity sequences, we illustrate how the addition of anchors can be used to modify alignments, while still maintaining the reproducibility and rigor of the rest of the alignment. Anchored alignments can be generated using the online version of AlignMe available at www.bioinfo.mpg.de/AlignMe/.

**Funding:** This research was supported by the Division of Intramural Research of the NIH, National Institute of Neurological Disorders and Stroke Z00-NS003143 (LRF; https://www.ninds.nih.gov/) and by the Max Planck Gesellschaft (LRF; https://www.mpg.de/de). The funders had no role in study design, data collection and analysis, decision to publish, or preparation of the manuscript.

**Competing interests:** The authors have declared that no competing interests exist.

## Introduction

Comparison of protein primary sequences is a fundamental step in analyzing evolutionary relationships and for accurate structural modeling based on homology [1, 2]. Solutions to the problem of primary sequence alignment typically focus either on identifying optimally-matched short fragments (local alignments) or on obtaining the best agreement between two entire sequences (global alignments). In both cases, the likelihood of matching of each pair of amino-acids in the alignment is scaled according to a reference table of amino-acid substitution rates, such as the BLOSUM, PAM, or VTML matrices [3–5], or in the case of alpha-helical membrane proteins, the JTT, PHAT, or SLIM matrices [6–8]. For a global alignment, the cost of introducing terminal gaps of different lengths must also be considered. From an algorithmic perspective, global alignments are commonly achieved using the Needleman-Wunsch algorithm [9] or hidden Markov Models [10, 11]. For both approaches, when the overall similarity of the two proteins is high, e.g., with > 50% identical residues, achieving accurate global alignments is relatively trivial [12]. At lower levels of similarity, accuracy can be improved by incorporating representations of properties such as secondary structure [13–17] or, in the specific case of membrane proteins, the membrane-spanning regions [18–20].

Even with the aforementioned advances, membrane proteins continue to present a significant challenge to pairwise sequence alignment methods, for two reasons. First, the regions that span the hydrophobic core of the lipid bilayer are highly hydrophobic and therefore their composition tends to be low in complexity [21]. Second, membrane proteins are thought to be among the most ancient of all proteins [22], and consequently the relationships between membrane proteins–even those with similar structural folds–can be very weak. The extent of this problem is reflected in the fact that even state-of-the-art methods such as PRALINE™ [18], HHalign [23], TM-Coffee [24] and AlignMe [19], fail to correctly align large fractions of the positions in the most dissimilar pairs of sequences in standard membrane protein test sets [25].

Fortunately, in real-life studies of protein structure and function, it is frequently the case that biochemical or biophysical data are available that identify a relationship between specific positions in the sequences being aligned. Examples might include the sites for substrate binding (see, e.g. **Fig 1**) or post-translational modifications. However, unless these residues belong to a larger sequence motif, their signal may be lost among the dissimilarities, and as a result, such positions may not be matched, rendering the final alignment uninformative. A typical solution is to make *ad hoc* manual modifications to the alignment. However, such interventions also inevitably modify the segments either side of the positions being matched, requiring arbitrary decisions regarding the boundaries and alignment of those neighboring segments, which reduces the robustness and reproducibility of the alignment overall.

In the absence of a 100% accurate alignment algorithm, an alternative strategy lies in the use of restraints, also referred to as anchors. Such restraints were first introduced in the context of the multiple-sequence alignment methods DIALIGN [26, 27] and COBALT [28], which use information from BLAST or CCD database searches to restrain fragments or domains during the process of constructing full-length protein or genome alignments. Other methods, such as MA-PRALINE [29], ConBind [30], and FMALIGN [31] allow for matching of regular expressions or sequence motifs. However, motif-matching is of limited value when only a single residue is conserved, or if the conservation is weak. The pairwise profile alignment tool SALIGN [32] accounts for this scenario by allowing anchoring of individual positions rather than sequence elements. However, none of the aforementioned methods are designed specifically for pairwise alignments of membrane proteins.

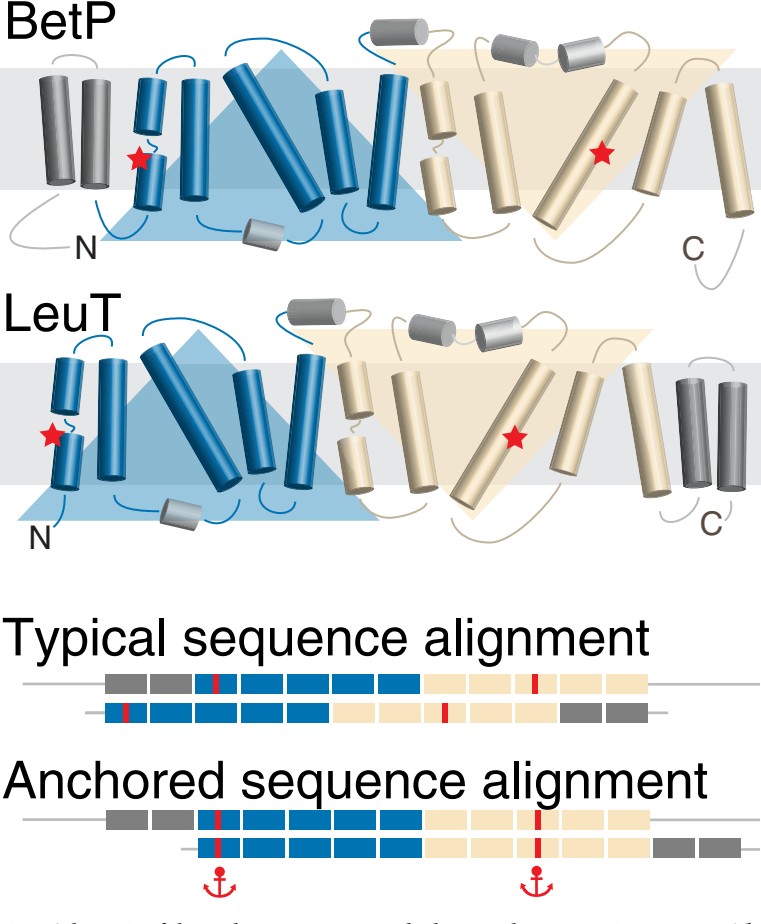

**Fig 1. Schematic of the anchors concept as applied to membrane protein sequences.** The example is for two proteins with the so-called LeuT fold, which comprises two repeats of five membrane-spanning segments each (*blue and wheat*). The protein BetP differs from LeuT in that it contains two additional transmembrane segments (*dark gray*) at the N-terminal end, rather than at the C-terminal end of the repeats. A typical sequence alignment favors matching of all twelve transmembrane segments and thus incorrectly aligns the core segments, including the position of the sodium binding sites (*red*). These errors can be corrected by using anchors at specific locations, here illustrated by the example of two residues in each protein known, based on biochemical studies, to bind a sodium ion.

Here, we introduce restraints as a feature in our method AlignMe, which was developed for pairwise global alignment of helical membrane protein sequences [19, 33]. The specificity for membrane proteins was incorporated into AlignMe in two stages of its development. First, the accuracy of alignments obtained with different types of input information, namely sequence homologs, hydrophobicity profiles, secondary structure, and transmembrane predictions, was assessed for a data set of membrane proteins [19]; the optimal combination of inputs was found to depend on the similarity level of the sequences involved. Second, the choice of gap penalties was optimized independently for each combination of inputs [19, 33]. Consequently, in practice, AlignMe is a suite of alignment "modes" tailored to the nature of the input sequences [19, 33].

The purpose of the anchors feature is to allow reproducible, user-guided modification of AlignMe alignments, independent of the mode of usage. The selection of positions to be anchored could be informed by data from any number of sources. For example, biochemical studies can provide evidence of the importance of specific amino acids during function, while spectroscopic data can indicate the role of specific residues in interactions with other cellular

components such as lipid headgroups. Here, we illustrate the application of such data to AlignMe alignments, using small numbers of anchors to align short peptides or long transmembrane proteins. For bioinformatic studies, we provide a further example, reflective of a situation in which full-length alignments of two proteins are required, while structural data, or local alignment methods, provide robust information for partial segments of the alignment, similar to the situations targeted by DIALIGN and COBALT. Together, these examples show how AlignMe anchoring contributes to the toolbox for membrane proteins sequence alignment methods.

## Materials and methods

### Anchoring

The Needleman-Wunsch global pairwise alignment algorithm [9] requires the generation of a so-called dynamic programming matrix that reflects the scores of matching every pair of residues in the two sequences based on a scoring function of pre-defined nature. For two sequences of lengths $N$ and $M$, this matrix has dimensions ($N$+1) x ($M$+1). The optimal alignment is ultimately selected as the highest-scoring path obtained by tracing a line from the lower-right corner (the C-terminal end) back to the top-left corner (the N-terminal end) of the matrix.

The ability to anchor positions in the alignment is implemented in AlignMe as follows. First, the dynamic programming matrix is initialized by placing non-zero values, of weight W, in all cells in the lower left and upper right quadrants of the matrix that are bounded by the row and column that we wish to anchor. The remainder of the cells are set to zero. The standard dynamic programming matrix is then constructed on top of this anchored matrix, by subtracting those non-zero values ($W$) in the regions defined by the anchored cell(s). As the traceback procedure follows the path with the highest scores, it will avoid regions that do not pass through the anchored positions, i.e., it will avoid alignments that do not match the pair of residues being fixed.

### SERCA regulatory peptide alignments

Seven different regulatory subunits of the calcium pump, including *Drosophila* sarcolamban isoform a, dSCLa (UNIPROT codes: C0HJH4, SLCA_DROME) and isoform b, dSCLb (M9PCQ8, M9PCQ8_DROME); human myoregulin, hMLN (P0DMT0, MLN_HUMAN); human phospholamban, hPLN (Q5R352, Q5R352_HUMAN), human sarcolipin, hSLN (O00631, SARCO_HUMAN); mouse another regulin, mALN (Q99M08, CD003_MOUSE); and mouse endoregulin, mELN (Q3U0I6, Q3U0I6_MOUSE), were aligned pairwise, using AlignMe in three different modes with default parameters for each mode [19, 33]: (1) Fast mode, which considers hydrophobicity and the VTML substitution matrix [5]; (2) AlignMe P mode, which considers a position-specific substitution matrix (PSSM) obtained from PSI-BLAST [34]; and (3) AlignMe PS mode, which considers the PSSM as well as secondary structure predictions from PSIPRED [35]. Note that the fourth available mode, AlignMe PST was not used for this test case. That is because the PST mode relies on transmembrane predictions from OCTOPUS [36], which did not consistently identify the transmembrane segment in all seven proteins; as such, not all pairs of alignments could be computed, preventing a comprehensive comparison.

Output alignments were converted to a single representative multiple-sequence alignment for visualization by merging the pairwise alignments obtained for hPLN using pyali v0.1.1 (https://github.com/christang/pyali). Alignments were visualized using Jalview v2.10.3b1 [37].

Anchors of $W = 1000$, a very high value chosen so as to force exact matching of the anchored positions, were imposed in each pairwise alignment at several positions in the transmembrane segments. The selection of these anchored positions and the reference alignments were taken from Primeau et al [38] and Anderson et al [39]. Test alignments were scored using the metrics described below, for the transmembrane domains. The transmembrane region was defined as residues 25–47 in hPLN, a conservative definition that avoided gaps in either of the two reference alignments.

## Multi-spanning membrane protein alignments

Alignments between *Corynebacterium glutamicum* BetP (UniProt entry L7V4D0) and *Aquifex aeolicus* LeuT (UniProt entry O67854) were obtained using AlignMe in PST mode. Residues 150 and 467 in BetP were anchored to residues 23 and 354 in LeuT, respectively. These positions, in the first and eighth transmembrane segments of the core LeuT fold, respectively, are known to form a conserved sodium ion binding site in both proteins, and in many other transporters of this superfamily [40, 41]. To be able to score the anchored alignments, we used as a reference a structure-based alignment of the two proteins [41] in which the core of the fold is superposed, as obtained previously using SKA [42] on PDB entries 2A65 and 4C7R, for LeuT and BetP, respectively, in addition to AdiC, ApcT, CaiT, Mhp1 and SGLT [41].

To assess the impact of different anchor strengths, the same alignments were recomputed by scaling $W$ over four orders of magnitude. The resultant alignments were assessed according to their similarity to that obtained with the highest tested weight ($W = 1000$) in which the anchored positions were confirmed to be matched.

## Alignments of the structural repeats of LeuT

Alignments were carried out between the repeated elements of LeuT, namely TM1-5 (residues 10–234) and TM6-10 (residues 242–443). The reference alignment was a structure-based alignment generated by the symmetry detection method, CE-Symm v2.0 [43, 44], and anchors were identified from that alignment. The full set of anchors covered the secondary structure elements, at residues 10–24, 26–37, 41–71, 76–85, 87–125, 136–153, 165–184, 190–214 and 223–232 of the first repeat. AlignMe was run in Fast mode [33], which would be a computationally-efficient choice for bioinformatic studies requiring large numbers of alignments.

**Measures of accuracy or agreement.** Absolute accuracy of a test alignment was measured by computing the number of positions in which the residues are identically matched as in the reference alignment. The number of accurate positions was then averaged over the length of the reference alignment.

The extent of the discrepancy between two alignments was measured using the so-called shift score, which is the number of columns by which a given residue is shifted from its expected position. This score was computed for each residue in the first sequence (when not aligned to a gap), and either plotted as a function of the position in the alignment or averaged over the number of non-gapped positions.

## Software and availability

The anchoring feature was implemented in AlignMe v1.2.2, and the code is available through GitHub (https://github.com/Lucy-Forrest-Lab/AlignMe). We also provide a web service at http://www.bioinfo.mpg.de/AlignMe/.

## Results

### Application of anchors to alignments of SERCA pump regulatory domains

Single-pass, or bitopic, membrane proteins can pose a significant challenge to a sequence alignment method, as their primary feature is a hydrophobic segment, around 25 amino acids in length. Nevertheless, even such simple proteins partake in specific functional interactions, as illustrated by the case of the peptides involved in regulating the **S**arco/**E**ndoplasmic **R**eticulum **C**alcium **A**TPase, also known as SERCA, the calcium pump, or $Ca^{2+}$-ATPase. These regulatory subunits, which include the well-studied examples phospholamban (PLN) and sarcolipin (SLN), modify the functional properties of SERCA through direct interactions involving their transmembrane subunits, as shown in structures of PLN and SLN in complex with SERCA [45–47]. Further modulation of those interactions is known, based on biochemical studies, to be mediated by residues on either side of the transmembrane region. Reports by Anderson et al [39] and Primeau et al [38] offer two alternative alignments for PLN and SLN with respect to several, more recently-identified, subunits in the same family. In the alignment by Primeau et al, the predicted transmembrane regions were used as a guide to match up segments, using a manual approach. Consequently, different residues were predicted to be forming the contacts with SERCA in the two studies. Moreover, in neither study were the regions outside of the transmembrane segment aligned.

As an illustration of the challenge posed by these sequences, pairwise alignments of seven known SERCA regulatory domains were first obtained without anchoring, using AlignMe in several different modes (Fast, P and PS; see Methods). The resultant alignments were then compared with those proposed in the two previous studies. Agreement was measured, for each pair of sequences, as the fraction of aligned positions in the transmembrane region that exactly match those in the published alignments. Clearly, many of the AlignMe alignments differ substantially from those proposed previously by either Primeau et al (**Fig 2B, orange)** or Anderson et al (**S1B Fig, orange**). Indeed, only around a third of the pairwise alignments agreed with those proposed by either Primeau et al [38] or Anderson et al [39] for >50% of the transmembrane region, while the remaining pairs differed along the full length of the transmembrane regions.

Since the above metric does not discriminate between alignments that are off by one position and those that differ by the entire length of a transmembrane segment, we also computed the mean shift score for each alignment. This metric indicates the number of positions by which a matched pair is offset from its expected position. The average was computed along the length of the transmembrane region. When no anchors were included in the alignments, the AlignMe alignments were shifted by anywhere up to 18 positions (**Fig 2C** and **S1C Fig, orange**), highlighting the challenge that these sequences present.

As mentioned above, several positions in PLN and SLN were used to guide the published alignments with the other regulatory domains, using a manual alignment approach. Here, we constructed alignments using AlignMe, by introducing anchors at those same positions, according to the match-ups proposed by Primeau et al [38] (**Fig 2A**) and Anderson et al [39] (**S1A Fig**). A very strong weight was imposed, essentially requiring that the anchored positions be fully matched (see Methods). We then asked whether the remainder of the alignment in the transmembrane region also agreed with that in the manual alignments, using the fraction of aligned transmembrane positions that agree with the reference (**Fig 2B** and **S1B Fig, cyan**), or the shift score (**Fig 2C** and **S1C Fig, cyan**). This analysis demonstrates that the anchors dramatically modify the alignments, such that all 21 pairwise alignments agree with the manually-constructed reference in over half of the positions (**Fig 2B** and **S1B Fig**). In addition, the extent of the shift error was dramatically reduced, with mean values of <1.5 for alignments obtained

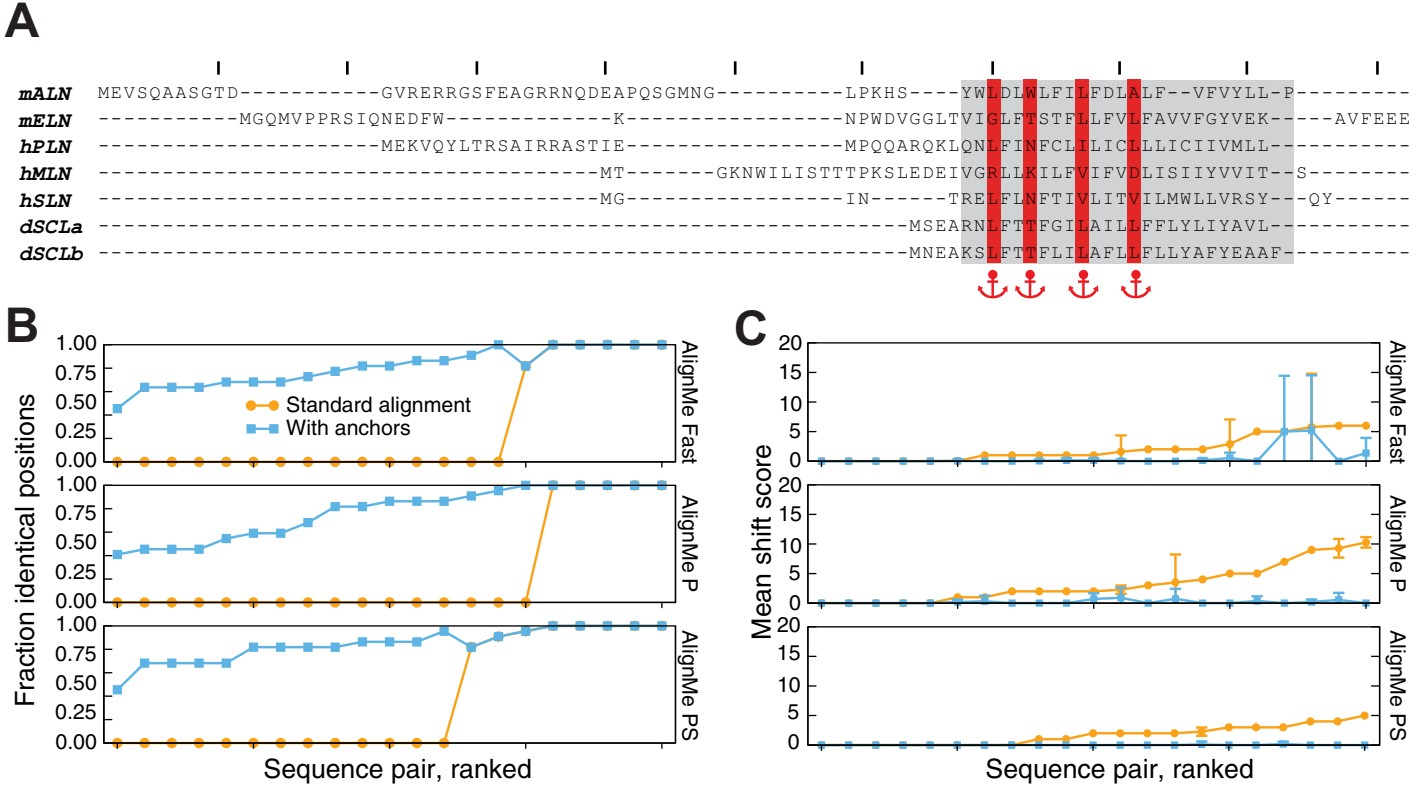

**Fig 2. Effect of applying anchors to pairwise alignments of SERCA regulatory subunits using AlignMe. (A)** Seven peptides, namely human phospholamban (hPLN), sarcolipin (hSLN), mouse another-regulin (mALN), mouse endoregulin (mELN), human myoregulin (hMLN), and *Drosophila* sarcolamban a and b isoforms (dSCLa & dSCLb) were pairwise aligned. Anchors were placed at the positions marked in red. In hPLN and hMLN, these residues are known to form contacts with SERCA; their equivalents in the other sequences were proposed previously [38]. Membrane-spanning segment definitions (*gray box*) are highlighted, and tick marks indicate every 10 positions along the alignment. **(B, C)** The effect of the anchors was assessed for pairwise alignments obtained either without (*orange circles*) or with (*cyan squares*) anchor constraints. The similarity of the AlignMe alignments to those proposed by Primeau et al [38], in the transmembrane region only, was measured as (B) the fraction of positions that are aligned identically, and (C) the extent of the shift discrepancy, or the number of positions by which each aligned column is shifted, relative to the position in the published alignment. The shift is measured as the mean over all residues in the transmembrane segments. Anchors were imposed at four positions in the transmembrane segment (*red in A*). Alignments were obtained using three different AlignMe modes: Fast mode (*upper*), which considers hydrophobicity and a standard substitution matrix; AlignMe P (*middle*), which considers a position-specific substitution matrix obtained from PSI-BLAST; and AlignMe PS (*lower*), which considers the PSSM as well as secondary structure predictions from PSIPRED. Pairs of sequences are sorted according to their accuracy obtained without anchoring. Error bars indicate standard deviations.

in PS mode, indicating that, even for positions not precisely in accordance with the previously-reported alignment, AlignMe typically placed those residues within 2 or three columns of their positions in the reference alignment (**Fig 2C** and **S1C Fig**).

These results illustrate that, even when the anchored positions are matched, the surrounding regions respond to the underlying alignment algorithm, allowing, for example, for gaps in the neighboring regions (see e.g., mALN in **Fig 2A**). Consequently, these alterations allow for a more robust prediction of the relationships between the residues not being forced to match. Moreover, it should be noted that more gaps were introduced in the transmembrane region when anchoring the positions proposed to match by Primeau et al, relative to those proposed by Anderson et al, suggesting that the latter might be more compatible with the results from the AlignMe method. Looking beyond the differences between the two reference methods though, we emphasize that, unlike a manual approach, anchoring allows the regions away from the anchored positions to be matched according to the underlying Needleman-Wunsch algorithm. This allows for new, testable predictions, e.g., if the phosphorylation sites known for hPLN are conserved in other regulatory domains.

**Anchoring binding site residues in alignments of LeuT fold transporters.**    While challenging, short peptides are not representative of a common challenge in the membrane protein field, namely mechanistic studies of large proteins with multiple membrane-spanning segments, whose highly hydrophobic, low-complexity transmembrane segments can be difficult to differentiate. We therefore also illustrate the application of anchors on another system, namely proteins belonging to the APC superfamily of transporters, which adopt the so-called LeuT fold (see **Fig 1**). In the case of the proteins LeuT and BetP, biochemical and biophysical experiments have demonstrated a common functional property of these proteins, namely a conserved sodium ion binding site involving side-chain and backbone groups in the first and eighth transmembrane segments of the core fold. Notably, in this case, high-resolution structures are available for BetP and LeuT that can be used to construct a reference for assessing the anchored alignments.

We first show that a standard alignment approach, even using the advanced features of AlignMe PST, does not even match the transmembrane segments in the two repeats (**Fig 3A**; blue and wheat for the first and second repeats, respectively), let alone the sodium binding site residues. Quantifying this result, the alignment is wildly incorrect for the entire length of the protein, with every pair of residues shifted by >200 from their positions in the reference alignment, as shown using the shift score (**Fig 4A**, orange). This result highlights the challenge of obtaining a global alignment for two proteins with highly dissimilar sequences and with multiple underlying motifs of hydrophobic elements. By contrast, imposing constraints at one or two positions in the known sodium ion binding site allows AlignMe to correctly match the core transmembrane segments. That is, segments in the first repeat of LeuT match with segments in the first repeat of BetP in alignments obtained with a single anchor closer to the beginning (**Fig 3B**) or the end (**Fig 3C**) of the proteins. Quantifying this result with the shift score shows that the majority of the positions in the >450-residue long alignment are shifted by less than five positions from the locations expected from the reference structure alignment (**Fig 4A**). Moreover, these improvements were achieved independent of whether the constraint was located toward the N-terminal end of the sequence, in TM1 (**Fig 4A, cyan circles**), toward the C-terminal end of the sequence, in TM8 (**Fig 4A, blue circles**), or in both positions (**Fig 4A, black squares**).

## The effect of modifying the anchor strength

To apply anchors in AlignMe, the only parameter that the user needs to provide, aside from the position of the anchor, is the weight, $W$, or strength, with which the anchor will be imposed. This value is effectively the bias that is applied to the dynamic programming matrix and influences the likelihood that the traceback algorithm will select an alignment that passes through this position in the matrix. In other words, the value of $W$ controls whether the final alignment will match the two residues involved; zero or small values of $W$ will not influence the traceback procedure, while infinitely high values would ensure that the alignment must match these positions. In the tests described above, we used a value of 1000, which ensured satisfaction of the anchored anchor constraints. However, it was not clear what the minimum value of $W$ would be in order to impose position matching. To assess the range of values at which an anchor starts to become influential on an alignment, we recomputed the BetP-LeuT pairwise alignments for a wide range of different values of $W$. We scored each alignment by the fraction of positions that are identical to that obtained with $W = 1000$ (**Fig 4B**). The results demonstrate that the obtained BetP-LeuT alignment is identical for all values of $W > 10$. However, with $W < 2$, the imposition of the restraint is not guaranteed. In that range, other factors, such as the local similarity between the sequences, may affect the alignment. We therefore

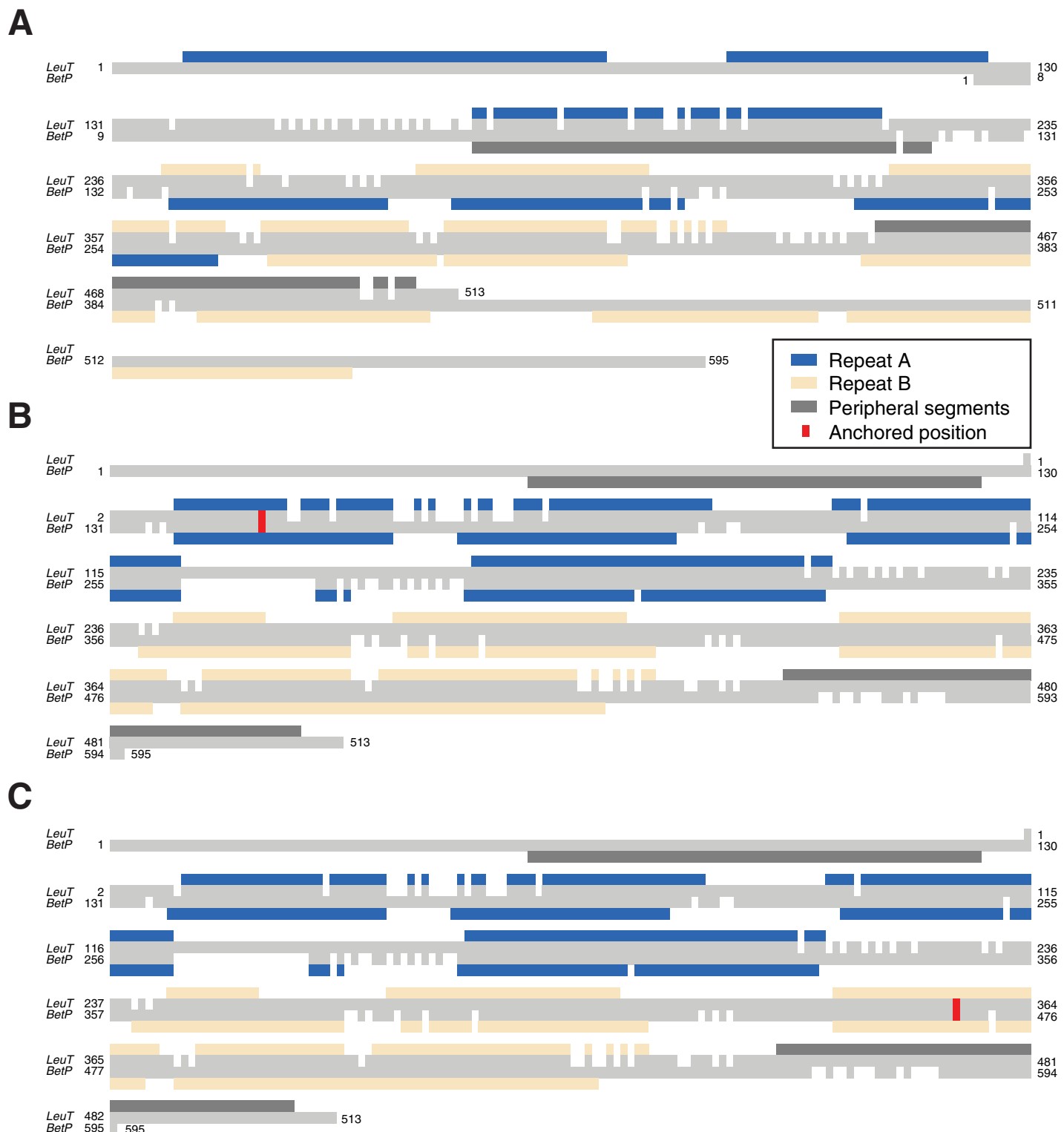

**Fig 3. Pairwise sequence alignments of the membrane proteins BetP and LeuT can be corrected using anchors.** Alignments were obtained either with a standard protocol (**A**) or with a single anchor imposed (*red*) restraining one column in either the first (**B**) or eighth (**C**) transmembrane segments of the core LeuT fold. Boxes above and below the respective sequences indicate the location of the transmembrane segments of the two proteins and highlight the related elements in the first (*blue*) and second (*wheat*) repeats. Transmembrane segments that are peripheral to the core fold are colored dark gray. See Fig 1 for more information.

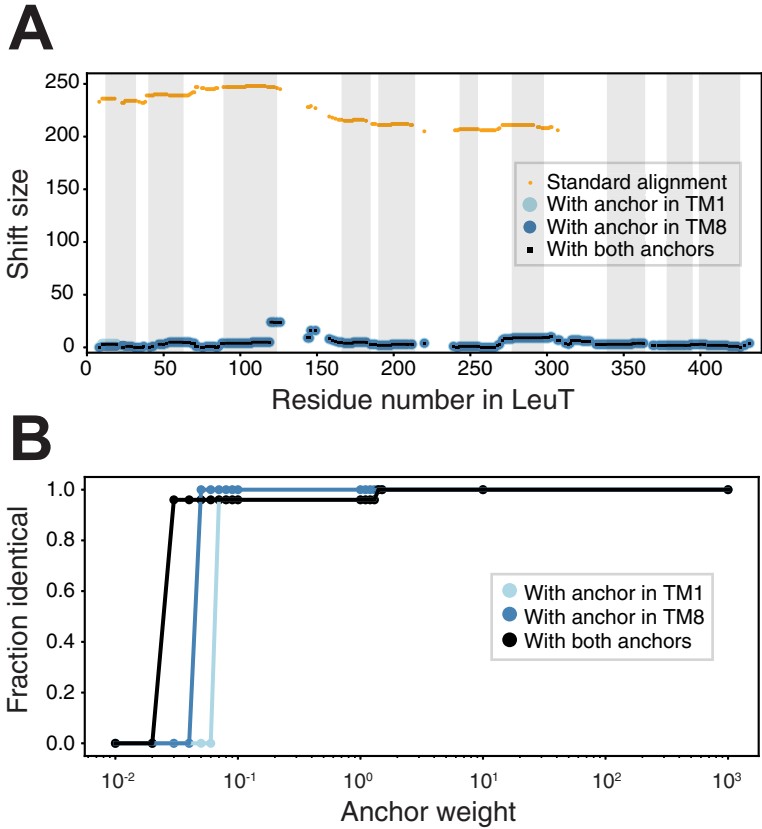

**Fig 4. The effect of anchors on alignments of the membrane proteins BetP and LeuT. (A)** The shift discrepancy, relative to a reference structure-based alignment, is plotted for each residue in LeuT, for alignments obtained either with the standard methodology (*orange*) or with anchors imposed (*light blue, dark blue or black*). The anchors are designed to target individual sodium-coordinating residues in the first or eighth transmembrane segment of the core LeuT fold, either separately (*light and dark blue*) or simultaneously (*black*). The approximate transmembrane regions of LeuT are indicated by gray boxes. Note that residues aligned to gaps are not scored. **(B)** Dependence of the alignment on the weight of the anchor, $W$, measured as the fraction of positions that are the same as in the alignment obtained using $W = 1000$. Alignments obtained with $W > 2$ are the same as those obtained with $W = 1000$, for all combinations of the two anchors. Weights smaller than 2 are not sufficient to overcome the contributions of other components of the similarity scores.

recommend that the user first applies a very high value of $W$, and checks to ensure that the desired positions have been matched in the resultant alignment. At that point, lower values of $W$ may be tested for their influence on the alignment.

## Anchors for ensuring accuracy in fragments of alignments

The above examples illustrate how just a few anchors can dramatically improve pairwise alignments of distantly related proteins, and therefore could be insightful for mechanistic and evolutionary studies of their function, or beneficial for structural modeling studies. Another practical application would include the need for a full-length alignment of two sequences, but where only fragments of the alignment are available. The sources of such aligned fragments might include high-quality local alignments, or structural alignments of repeated elements within a structure, as in the case of membrane-protein symmetry analysis [48, 49]. In such cases, it is desirable to have an unbiased, but computationally efficient, method by which to construct the remainder of the alignment while maintaining the exact matching for the fragment.

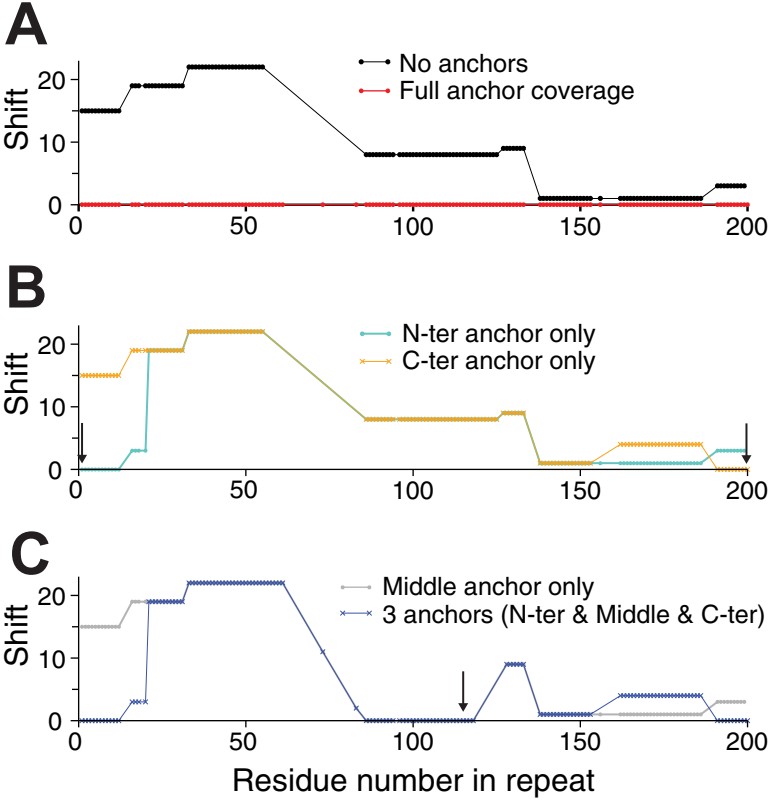

**Fig 5. Extensive anchor coverage guarantees accurate alignments when using computationally-efficient alignment methods for very challenging chases, shown for the internal repeats of LeuT aligned using AlignMe Fast mode.** The number of columns by which a residue is shifted from position in the reference alignment is shown for all aligned residues in the repeats. The reference alignment, obtained from a structure-based symmetry detection method, is shown in S2 Fig. Alignments were obtained **(A)** without any anchors (*black circles*), or with anchors covering all secondary structure elements (*red circles*), **(B)** with a single anchor at the N- or C-terminal end of the alignment (*cyan circles*, *orange crosses*, *respectively*), marked with arrows, **(C)** with a single anchor in the center of the alignment (*gray circles*), marked by an arrow, or with three anchors at the beginning, middle and end (*blue crosses*). Note that residues aligned to gaps are not scored using this measure, and the residue numbering corresponds to that in the second repeat.

As an illustration of such a situation, we examined the case of the internal repeats in LeuT, whose sequences contain <10% identical residues [50] (**S2 Fig**). Specifically, we aligned the two five-transmembrane-helix repeats (see **Fig 1**) which are known to be related by pseudo-symmetry (**Fig 5**). Since bioinformatic analyses are likely to involve many thousands or millions of alignments, we used the Fast mode of AlignMe to align these repeats. However, a standard alignment generated using AlignMe without any anchors, only contains 9.4% of its positions correctly aligned. In addition, the shift error is quite large along the length of the alignment (**Fig 5A, black**), highlighting the challenge of aligning this pair of sequences.

We then asked whether anchors would be able to maintain the matching in the repeated elements for this pair of sequences. The reference alignments used to define the anchors, and against which the accuracy of the alignment could be assessed, were obtained using a symmetry-detection algorithm, CE-Symm. A very strong weight ($W = 1000$) was used to ensure matching. Using all these symmetry-based restraints, all the residues were correctly aligned (**Fig 5A, red**).

We also asked whether complete coverage would be necessary, or if, like for the examples above, a handful of restraints would suffice. We therefore repeated the alignments using a

single restraint at either the N- or C-terminal end of the two sequences (**Fig 5B**). Each restraint was able to modify the alignment locally, as shown by a dramatic decrease in the shift score in the ends of the alignments, but the remainder of the alignment was unaffected. A single restraint in the middle of the alignment led to a similar, local improvement (**Fig 5C, gray**). Moreover, combining the three restraints to cover the entire alignment was also insufficient (**Fig 5C, blue**).

Together with the data for BetP and LeuT, these results indicate that, for bioinformatic analyses involving very distantly-related sequences, it will be important to anchor every position in the known fragment, to fully ensure matching across the entire segment.

## Discussion

In this work, we describe the implementation and application of an alignment anchoring approach in the AlignMe software package, as an additional tool for addressing the challenges of sequence alignment of membrane proteins. We note that anchors can, in principle, be applied to the alignment of any pair of primary sequences within the framework of a Needleman-Wunsch algorithm. Indeed, since the framework of the AlignMe software is such that it can read any type of input information or set of gap penalties, AlignMe itself can readily be repurposed for anchoring in any pair of sequences, including beta-barrel membrane proteins, or water-soluble proteins, simply by using standard position-specific substitution matrices or general substitution matrices such as BLOSUM.

The example of BetP and LeuT illustrates a situation involving large multi-spanning proteins, where the presence of two peripheral segments in different positions can stymie even the most robust of global alignment methods. In that case, a single constraint has the power to dramatically improve the alignments, sufficient for, e.g., a molecular modeling project.

In addition to functional studies of specific proteins, anchors may have a number of other applications in bioinformatic workflows. As mentioned above, they can be used to expand a robust, local alignment, such as that obtained from a structure-based, or motif-matching approach, into a full-length alignment suitable for situations where the entire protein sequence is required [27, 28]. In future work, for example, we plan to utilize the anchoring strategy in AlignMe in completing global alignments of internal structural repeats obtained by structure-based symmetry detection methods such as CE-Symm, so that those alignments can be readily compared across structures and family members of membrane proteins in the EncoMPASS database [48].

In the applications provided here, the strength of the anchors imposed was deliberately set to a very large number to ensure that they would override all other paths in the dynamic programming matrix, and thereby essentially fix the alignment at that point. This choice assumes that the two anchored positions have a strong evolutionary relationship. In situations where that relationship is not present, anchors might impose an unreasonable bias on the alignment. We therefore recommend that the user focuses on anchoring pairs of residues for which the evidence is strongest. Moreover, it is wise to compare alignments obtained with and without the anchors, and perhaps even to try scaling down the anchor strength to assess the effects of the anchor strength on the resultant alignment.

Finally, one might also imagine situations in which candidate anchors are mutually exclusive. AlignMe currently carries out error checking to prevent mutually incompatible anchors from being imposed during the same alignment run. However, in some scenarios it might be beneficial to allow the violation of one or more anchors; this possibility will be addressed in future work.

In conclusion, anchoring is a valuable addition to the arsenal of sequence alignment approaches, and its implementation within AlignMe makes it readily available to those interested in carrying out robust, reliable alignments of membrane proteins.

## Supporting information

**S1 Fig. Effect of applying anchors from Anderson et al to pairwise alignments of SERCA regulatory domains using AlignMe. (A)** Seven peptides, namely human phospholamban (hPLN), sarcolipin (hSLN), mouse another-regulin (mALN), mouse endoregulin (mELN), human myoregulin (hMLN), and *Drosophila* sarcolamban a and b isoforms (dSCLa & dSCLb) were aligned. Anchors were placed at the positions marked in red. In hPLN and hMLN, two of these residues form contacts with SERCA; the conservation of these positions in the other sequences was noted previously [39]. Membrane-spanning segment definitions (*gray box*) are highlighted, and tick marks indicate every 10 positions along the alignment. **(B, C)** The effect of the anchors was assessed for pairwise alignments obtained either without (*orange circles*) or with (*cyan squares*) anchor constraints. The similarity of the AlignMe alignments to those proposed by Anderson et al, in the transmembrane regions only, was measured as (B) the fraction of positions that are aligned identically, and (C) the extent of the shift discrepancy, or the number of positions by which each aligned column is shifted. The shift is measured relative to the positions in the published alignment, and the mean shift is an average over all residues in the transmembrane segments. Anchors were imposed at six positions in the transmembrane segment (*red in A*). Alignments were obtained using three different AlignMe modes: fast mode (*upper*), which considers hydrophobicity and a standard substitution matrix; AlignMe P (*middle*), which considers a position-specific substitution matrix obtained from PSI-BLAST; and AlignMe PS (*lower*), which considers the PSSM as well as secondary structure predictions from PSIPRED. Pairs of sequences are sorted according to their accuracy obtained without anchoring. Error bars indicate standard deviations.
(EPS)

**S2 Fig. Pairwise alignment of the repeats of LeuT obtained using the symmetry-detection algorithm CE-Symm, used as a reference for the pairwise alignment accuracy and to select anchors.** Red boxes indicate the positions of anchors implemented in Fig 5. Identical residues are colored dark blue. The location of two residues not resolved in the structure of LeuT is indicated with a vertical line. Figure was generated using Jalview.
(EPS)

## Acknowledgments

We thank Drs. Mykola Petrov and Markus Rampp for their support of the MPG supercomputing center and the AlignMe server.

## Author Contributions

**Conceptualization:** René Staritzbichler, Antoniya Aleksandrova, Kamil Khafizov, Lucy R. Forrest.

**Data curation:** Edoardo Sarti, Emily Yaklich, Lucy R. Forrest.

**Formal analysis:** Edoardo Sarti, Emily Yaklich, Lucy R. Forrest.

**Funding acquisition:** Lucy R. Forrest.

**Methodology:** René Staritzbichler, Edoardo Sarti, Antoniya Aleksandrova.

**Resources:** Lucy R. Forrest.

**Software:** René Staritzbichler, Edoardo Sarti, Emily Yaklich, Marcus Stamm.

**Supervision:** René Staritzbichler, Antoniya Aleksandrova, Lucy R. Forrest.

**Validation:** Edoardo Sarti, Lucy R. Forrest.

**Visualization:** Emily Yaklich, Lucy R. Forrest.

**Writing – original draft:** René Staritzbichler, Kamil Khafizov, Lucy R. Forrest.

**Writing – review & editing:** René Staritzbichler, Edoardo Sarti, Emily Yaklich, Antoniya Aleksandrova, Marcus Stamm, Kamil Khafizov, Lucy R. Forrest.

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
