## [Decision Letter · Decision Letter 0]

6 Oct 2020

PONE-D-20-28678

Refining pairwise sequence alignments of membrane proteins by the incorporation of anchors

PLOS ONE

Dear Dr. Forrest

Thank you for submitting your manuscript to PLOS ONE. After careful consideration, we feel that it has merit but does not fully meet PLOS ONE’s publication criteria as it currently stands. Therefore, we invite you to submit a revised version of the manuscript that addresses the points raised during the review process.

All reviewers unanimously found your contribution interesting but limited and requested explanations as well as further demonstration of the added value of the software. Please expand your arguments and justify better the applicability of the method.

We look forward to receiving your revised manuscript.

Kind regards,

Frederique Lisacek

Academic Editor

PLOS ONE

Journal Requirements:

Reviewers' comments:

Reviewer's Responses to Questions

**Comments to the Author**

1. Is the manuscript technically sound, and do the data support the conclusions?

Reviewer #1: Partly

Reviewer #2: Yes

Reviewer #3: Partly

2. Has the statistical analysis been performed appropriately and rigorously? 

Reviewer #1: N/A

Reviewer #2: I Don't Know

Reviewer #3: Yes

3. Have the authors made all data underlying the findings in their manuscript fully available?

Reviewer #1: No

Reviewer #2: Yes

Reviewer #3: Yes

4. Is the manuscript presented in an intelligible fashion and written in standard English?

Reviewer #1: Yes

Reviewer #2: Yes

Reviewer #3: Yes

5. Review Comments to the Author

Reviewer #1: The authors describe a new feature that they implemented in the AlignMe software, the anchoring of residues, to improve pairwise alignments of membrane proteins of remote homology.

The manuscript is well written, clearly presented and interesting. The functionality in itself has merit.

I have the following comments regarding the manuscript:

- My understanding is that anchoring isn't new, and that the novelty described in this paper is the implementation of such an algorithm in AlignMe.

The introduction is extensive and interesting for a general audience, but nearly gives the impression that anchoring is the novelty described in this manuscript. So do the results, which are interesting to see, but don't show any novel insight on the protein sequences taken as input (the performance of AlignMe being the novelty). The authors should clarify the scope of the study.

- The introduction should describe ways anchors can be determined. My understanding is that all the anchors used in this manuscript are derived from structural data. Is this true in general, or are there other ways one can do it?

Can the authors

- In terms of reproducibility, I don't see how anchors fundamentally differ from manual adjustments. In the end, one leverages external evidence not visible from the sequence itself to refine the alignment. Can the authors clarify exactly how they think anchors improve reproducibility over manual adjustments?

- The authors claim that "new, anchored alignments, [...] could potentially contain mechanistic insights [...]". However if my understanding that anchors are mainly determined from structural evidence is correct, I don't see what additional insight the alignment could provide, that wouldn't have been visible from the experimental evidence itself.

- Clearly anchoring provides significant improvements over naive pairwise alignment methods. However we live in the age of deep sequencing. Tools such as HH-suite proved very effective at identifying even remote homology in a fully automated manner. They can accurately identify conserved residues even in very remote homology cases without the need for manual intervention. It would be relevant to compare the results of anchored pairwise alignments with more challenging methods like this, for instance using pairwise HHMs with sequence profiles built on deep MSAs. Does anchoring still provide more accurate alignments in this scenario?

In addition AlignMe has a profile-profile mode. Why was this not used as a comparison point? It should be straightforward to obtain deep MSAs for all the sequences used in this paper...

This should be doable for both the leuT and SERCA examples.

- In addition, the authors may want to describe how to use this feature in practice in AlignMe. As one needs to define custom alignment parameters to add anchors, it would be useful to provide some guidance on how to set the various parameters. This might also help clarify the scope of this article.

- The authors write "The above examples are limited by lack of direct structural data", I believe referring to the SERCA pump. However there is a wealth of structural data described in Primeau 2018. Can the authors clarify this sentence?

- The authors write "First, optimal forms of input [...]". I don't understand this sentence. What does "Form of input" mean? How does it relate to the manuscript? This reads like a method but I don't see it actually addressed in the methods/results section. How is optimal defined, and what is optimized?

- The software doesn't have an open source license, which is against the PLOS ONE guidelines <https://journals.plos.org="" plosone="" s="" submission-guidelines="">. As the source code is already available on GitHub this should be easy to address.</https:>

Reviewer #2: Existing tools fail to properly align transmembrane proteins due to the large numbers of motif repeats. The authors propose a new approach for their AlignMe tool to improve the alignments of transmembrane proteins. It consists of specifying a pair of residues in the source and the target that must be aligned together, thus constraining the overall alignment. This can cause large offsets, from a default alignment, while still making the final alignment valid. This method is very interesting but requires precise knowledge of the secondary structure of the aligned sequences to define the anchors.

Major revisions:

Figure 4 has an axis that misrepresents the distinctions between results with TM1, TM8 or both anchors. Showing that the 20-250 residue shift is present over the entire alignment without the anchors is not very useful. In addition, using colored points is less precise than straight lines.

Minor revisions:

Specify in the abstract and in the intro that the AlignMe tool is available online. Specify in the intro that your team previously developed AlignMe and proposes a new criterion to improve this tool. It is not clear right now.

Why use the TM1 and TM8 anchors, rather than the others ?

Figure 3 lacks text annotations to specify the positions of the anchors and the meaning of the red lines of the anchors.

Figure 5 does not show the contribution of each anchor separately. The legend is not clear, it is not necessary to list all the names of the anchors when the count is sufficient. An addition in the figure or an additional table specifying the positions of the anchors and therefore of the shifts that can be corrected would be useful.

For all the figures, their markup in the text is brutal and does not allow a correct understanding of their usefulness without referring to the legends of the figures.

Discussion: ‘This is because global alignment methods favor …’ is redundant

This approach could be used to align other types of proteins made up of repeats, such as propellers. Are you planning to make AlignMe available in this type of case?

For the website: Add a more interactive interface to define the anchors with the possibility of launching a first analysis of the secondary structures to then select the appropriate anchors. Their placement in all the advanced criteria does not make them accessible.

Reviewer #3: Overall the manuscript is well written and makes an interesting potential contribution to PLoS One. The work presented is straightforward and somewhat simplistic. The authors could have tested their method and presented their results for additional test cases of increasing difficulty to make their conclusions more compelling. In their results, they only show two examples that work well with their algorithm, and they do not seem to address that increasing constraints may lead to user bias. Can an incorrect anchor lead to a seemingly correct alignment? Additionally, is this algorithm compatible with beta-barrel membrane proteins, which may be worth mentioning or investigating.

1. Page 6: Authors do make a point for the usefulness of anchoring residues in the alignment. But this is not new and has been used in manual approaches to sequence alignment. What are the criteria for particular residues to be selected for anchoring during alignment? The author’s use well-known examples, but potential anchor residues could be more ambiguous (residues that impact function, but the substrate binding residues are unknown). The impact of the weight used for anchor residues would also be informative.

2. Figure 2B & C: A better description of the y-axes would help non-expert readers.

3. Overall, the figures were poor quality (blurry/pixelated).

4. The alignment in Figure 2A is speculative and an alternative alignment has been presented (Anderson et al., Sci Signal 2017). It would have been informative to test both sets of anchoring residues and possibly discern criteria for “correct” versus “incorrect” alignments. The alignment of peptides presents unique challenges.

5. The cases presented in the results appear to be benchmarking cases. What is lacking is one or more unknown cases where sequence and evolutionary relationships are less clear, but functional data (important residues) are available.

6. Clearly, the use of anchors is interesting, but there is no attempt to characterize the quality of the anchor and the impact on AlignMe. For proteins where there is no structural information and mutagenesis data are available, what is the impact of using “fuzzy” anchors. For that matter, LeuT and BetP could have been compared using “fuzzy” anchors and lower weights to better balance the imposed anchor with the scoring path through the matrix.

6. PLOS authors have the option to publish the peer review history of their article (what does this mean?). If published, this will include your full peer review and any attached files.

Reviewer #1: No

Reviewer #2: No

Reviewer #3: No

---

## [Author Response · Author response to Decision Letter 0]

29 Mar 2021

Please see attached file named "responses to reviewers"

---

## [Decision Letter · Decision Letter 1]

15 Apr 2021

Refining pairwise sequence alignments of membrane proteins by the incorporation of anchors

PONE-D-20-28678R1

Dear Dr. Forrest,

We’re pleased to inform you that your manuscript has been judged scientifically suitable for publication and will be formally accepted for publication once it meets all outstanding technical requirements.

Kind regards,

Frederique Lisacek

Academic Editor

PLOS ONE

Additional Editor Comments (optional):

Reviewers' comments:

Reviewer's Responses to Questions

**Comments to the Author**

1. If the authors have adequately addressed your comments raised in a previous round of review and you feel that this manuscript is now acceptable for publication, you may indicate that here to bypass the “Comments to the Author” section, enter your conflict of interest statement in the “Confidential to Editor” section, and submit your "Accept" recommendation.

Reviewer #1: All comments have been addressed

Reviewer #2: All comments have been addressed

Reviewer #3: All comments have been addressed

2. Is the manuscript technically sound, and do the data support the conclusions?

Reviewer #1: Yes

Reviewer #2: Yes

Reviewer #3: (No Response)

3. Has the statistical analysis been performed appropriately and rigorously? 

Reviewer #1: Yes

Reviewer #2: Yes

Reviewer #3: (No Response)

4. Have the authors made all data underlying the findings in their manuscript fully available?

Reviewer #1: Yes

Reviewer #2: Yes

Reviewer #3: (No Response)

5. Is the manuscript presented in an intelligible fashion and written in standard English?

Reviewer #1: Yes

Reviewer #2: Yes

Reviewer #3: (No Response)

6. Review Comments to the Author

Reviewer #1: (No Response)

Reviewer #2: The authors have corrected the manuscript according to the different reviewers comments leading to large modification in the manuscript. The manuscript and associated work is now presented with a high quality.

Reviewer #3: (No Response)

7. PLOS authors have the option to publish the peer review history of their article (what does this mean?). If published, this will include your full peer review and any attached files.

Reviewer #1: No

Reviewer #2: No

Reviewer #3: No

---

## [Editor Report · Acceptance letter]

22 Apr 2021

PONE-D-20-28678R1 

Refining pairwise sequence alignments of membrane proteins
by the incorporation of anchors 

Dear Dr. Forrest:

I'm pleased to inform you that your manuscript has been deemed suitable for publication in PLOS ONE. Congratulations! Your manuscript is now with our production department. 

Kind regards, 

on behalf of

Dr. Frederique Lisacek 

Academic Editor

PLOS ONE